# Online Structure Learning for Feed-Forward and Recurrent Sum-Product Networks

**Agastya Kalra**,[*]   **Abdullah Rashwan**,[*]   **Wilson Hsu**,   **Pascal Poupart**
Cheriton School of Computer Science, Waterloo AI Institute, University of Waterloo, Canada
Vector Institute, Toronto, Canada
agastya.kalra@gmail.com,{arashwan,wwhsu,ppoupart}@uwaterloo.ca

**Prashant Doshi**
Department of Computer Science
University of Georgia, USA
pdoshi@cs.uga.edu

**George Trimponias**
Huawei Noah's Ark Lab, Hong Kong
g.trimponias@huawei.com

## Abstract

Sum-product networks have recently emerged as an attractive representation due to their dual view as a special type of deep neural network with clear semantics and a special type of probabilistic graphical model for which marginal inference is always tractable. These properties follow from the conditions of completeness and decomposability, which must be respected by the structure of the network. As a result, it is not easy to specify a valid sum-product network by hand and therefore structure learning techniques are typically used in practice. This paper describes a new *online* structure learning technique for feed-forward and recurrent SPNs. The algorithm is demonstrated on real-world datasets with continuous features and sequence datasets of varying length for which the best network architecture is not obvious.

## 1   Introduction

Sum-product networks (SPNs) were introduced as a new type of deep representation [13] equivalent to arithmetic circuits [3]. They distinguish themselves from other types of neural networks by several desirable properties: 1) The quantities computed by each node can be clearly interpreted as (un-normalized) probabilities. 2) SPNs can represent the same discrete distributions as Bayesian and Markov networks [19] while ensuring that exact inference[2] has linear complexity with respect to the size of the network. 3) SPNs are generative models that naturally handle arbitrary queries with missing data while allowing the inputs and outputs to vary.

There is a catch: these nice properties arise only when the structure of the network satisfies the conditions of decomposability and completeness [13]. Hence, it is not easy to specify sum-product networks by hand. In particular, fully connected networks typically violate those conditions. While this may seem like a major drawback, the benefit is that researchers have been forced to develop structure learning techniques to obtain valid SPNs that satisfy those conditions [4, 7, 12, 9, 16, 1, 18, 14, 10]. In deep learning, feature engineering has been replaced by architecture engineering, however this is a tedious process that many practitioners would like to automate. Hence, there is a need for scalable structure learning techniques.

---

[*]Equal contribution, first author was selected based on a coin flip
[2]Most types of inference are tractable except for marginal MAP inference, which is still intractable for SPNs.

To that effect, we propose a new *online* structure learning technique for feed-forward and recurrent SPNs [10]. The approach starts with a network structure that assumes that all features are independent. This network structure is then updated as a stream of data points is processed. Whenever a non-negligible correlation is detected between some features, the network structure is updated to capture this correlation. The approach is evaluated on several large benchmark datasets, including sequence data of varying length.

## 2    Background

Poon and Domingos presented SPNs as a new type of deep architecture consisting of a rooted acyclic directed graph with interior nodes that are sums and products while the leaves are tractable distributions, including Bernoulli distributions for discrete SPNs and Gaussian distributions for continuous SPNs. Each edge emanating from sum nodes is labeled with a non-negative weight $w$. An SPN encodes a function $f(\mathbf{X} = \mathbf{x})$ that takes as input a variable assignment $\mathbf{X} = \mathbf{x}$ and produces an output at its root. This function is defined recursively at each node $i$ as follows:

$$f_i(\mathbf{X} = \mathbf{x}) = \begin{cases} \Pr(\mathbf{X_i} = \mathbf{x_i}) & \text{if } isLeaf(i) \\ \sum_j w_j f_{child_j(i)}(\mathbf{x}) & \text{if } isSum(i) \\ \prod_j f_{child_j(i)}(\mathbf{x}) & \text{if } isProduct(i) \end{cases}$$

Here, $\mathbf{X_i} = \mathbf{x_i}$ denotes the variable assignment restricted to the variables contained in the leaf $i$. If none of the variables in leaf $i$ are instantiated by $\mathbf{X} = \mathbf{x}$ then $\Pr(\mathbf{X_i} = \mathbf{x_i}) = \Pr(\emptyset) = \mathbf{1}$. If leaf $i$ contains continuous variables, then $\Pr(\mathbf{X_i} = \mathbf{x_i})$ should be interpreted as $pdf(X_i = x_i)$.

An SPN is a neural network in the sense that each interior node can be interpreted as computing a linear combination (for sum node) or non-linear combination (for product node) of its children. An SPN can also be viewed as encoding a joint distribution over the random variables in its leaves when the network structure satisfies certain conditions. These conditions are often defined in terms of the notion of *scope*.

**Definition 1** (Scope). *The $scope(i)$ of a node $i$ is the set of variables that are descendants of $i$.*

A sufficient set of conditions to ensure that the SPN encodes a valid joint distribution includes:

**Definition 2** (Completeness [2, 13]). *An SPN is complete if all children of the same sum node have the same scope.*

**Definition 3** (Decomposability [2, 13]). *An SPN is decomposable if all children of the same product node have disjoint scopes.*

Decomposability allows us to interpret product nodes as computing factored distributions with respect to disjoint sets of variables, which ensures that the product is a valid distribution over the union of the scopes of the children. Similarly, completeness allows us to interpret sum nodes as computing a mixture of the distributions encoded by the children since they all have the same scope. Each child is a mixture component with mixture probability proportional to its weight. Hence, in complete and decomposable SPNs, the sub-SPN rooted at each node can be interpreted as encoding an unnormalized joint distribution over its scope. We can use the function $f$ to answer inference queries with respect to the joint distribution encoded by the entire SPN: Marginal queries: $\Pr(\mathbf{X} = \mathbf{x}) = \frac{\mathbf{f_{root}}(\mathbf{X=x})}{\mathbf{f_{root}}(\emptyset)}$; conditional queries: $\Pr(\mathbf{X=x}|\mathbf{Y=y}) = \frac{\mathbf{f_{root}}(\mathbf{X=x,Y=y})}{\mathbf{f_{root}}(\mathbf{Y=y})}$.

Unlike most neural networks that can answer queries with fixed inputs and outputs only, SPNs can answer conditional inference queries with varying inputs and outputs simply by changing the set of variables that are queried (outputs) and conditioned on (inputs). Furthermore, SPNs can be used to generate data by sampling from the joint distributions they encode. This is achieved by a top-down pass through the network. Starting at the root, each child of a product node is followed, a single child of a sum node is sampled according to the unnormalized distribution encoded by the weights of the sum node and a variable assignment is sampled in each leaf that is reached. This is particularly useful in natural language generation tasks and image completion tasks [13].

Note also that inference queries other than marginal MAP can be answered exactly in linear time with respect to the size of the network since each query requires two evaluations of the network function $f$ and each evaluation is performed in a bottom-up pass through the network. This means that SPNs can

also be viewed as a special type of tractable probabilistic graphical model, in contrast to Bayesian and Markov networks for which inference is #P-hard [17]. Any SPN can be converted into an equivalent bipartite Bayesian network without any exponential blow up, while Bayesian and Markov networks can be converted into equivalent SPNs at the risk of an exponential blow up [19]. In practice, we do not convert probabilistic graphical models (PGMs) into SPNs since we typically learn SPNs directly from data. The tractable nature of SPNs ensures that the resulting distribution permits exact tractable inference.

Melibari et al. [10] proposed dynamic SPNs (a.k.a. recurrent SPNs) to model sequence data of variable length. A recurrent SPN consists of a bottom network that feeds into a template network (repeated as many times as needed) that feeds into a top network. The template network describes the recurrent part of the network. Inputs to the template network include data features and interface nodes with the earlier part of the network while the output consists of nodes that interface with the previous and subsequent part of the network. Melibari et al. [10] describe an invariance property for template networks that ensures that the resulting recurrent SPN encodes a valid distribution.

## 2.1 Parameter Learning

The weights of an SPN are its parameters. They can be estimated by maximizing the likelihood of a dataset (generative training) [13] or the conditional likelihood of some output features given some input features (discriminative training) by stochastic gradient descent (SGD) [6]. Since SPNs are generative probabilistic models where the sum nodes can be interpreted as hidden variables that induce a mixture, the parameters can also be estimated by the Expectation Maximization schema (EM) [13, 11]. Zhao et al. [21] provides a unifying framework that explains how likelihood maximization in SPNs corresponds to a signomial optimization problem where SGD is a first order procedure, sequential monomial approximations are also possible and EM corresponds to a concave-convex procedure that converges faster than other techniques. Since SPNs are deep architectures, SGD and EM suffer from vanishing updates and therefore "hard" variants have been proposed to remedy this problem [13, 6]. By replacing all sum nodes by max nodes in an SPN, we obtain a max-product network where the gradient is constant (hard SGD) and latent variables become deterministic (hard EM). It is also possible to train SPNs in an online fashion based on streaming data [9, 15, 20, 8]. In particular, it was shown that online Bayesian moment matching [15, 8] and online collapsed variational Bayes [20] perform much better than SGD and online EM.

## 2.2 Structure Learning

Since it is difficult to specify network structures for SPNs that satisfy the decomposability and completeness properties, several automated structure learning techniques have been proposed [4, 7, 12, 9, 16, 1, 18, 14, 10]. The first two structure learning techniques [4, 7] are top down approaches that alternate between instance clustering to construct sum nodes and variable partitioning to construct product nodes. We can also combine instance clustering and variable partitioning in one step with a rank-one submatrix extraction by performing a singular value decomposition [1]. Alternatively, we can learn the structure of SPNs in a bottom-up fashion by incrementally clustering correlated variables [12]. These algorithms all learn SPNs with a tree structure and univariate leaves. It is possible to learn SPNs with multivariate leaves by using a hybrid technique that learns an SPN in a top down fashion, but stops early and constructs multivariate leaves by fitting a tractable probabilistic graphical model over the variables in each leaf [16, 18]. It is also possible to merge similar subtrees into directed acyclic graphs in a post-processing step to reduce the size of the resulting SPN [14].

In the context of recurrent SPNs, Melibari et al. [10] describe a search-and-score structure learning technique that does a local search over the space of template network structures while using scoring based on log likelihood computations.

So far, all these structure learning algorithms are batch techniques that assume that the full dataset is available and can be scanned multiple times. Lee et al. [9] describe an online structure learning technique that gradually grows a network structure based on mini-batches. The algorithm is a variant of LearnSPN [7] where the clustering step is modified to use online clustering. As a result, sum nodes can be extended with more children when the algorithm encounters a mini-batch that exhibits additional clusters. Product nodes are never modified after their creation. This technique requires

large mini-batches to detect the emergence of new clusters and it assumes fixed length data so it is unable to generate structures for recurrent SPNs.

In this paper, we describe the first *online* structure learning technique for *feed-forward and recurrent* SPNs. It is more accurate and it scales better than the offline search-and-score technique introduced previously [10]. It also scales better than the technique that uses online clustering [9] while working with small mini-batches and recurrent SPNs.

# 3   Online Learning

To simplify the exposition, we assume that the leaf nodes have Gaussian distributions (though we show results in the experiments with Bernoulli distributions and it is straightforward to generalize to other distributions). A leaf node may have more than one variable in its scope, in which case it follows a multivariate Gaussian distribution. Suppose we want to model a probability distribution over a $d$-dimensional space. The algorithm starts with a fully factorized joint probability distribution over all variables, $p(\mathbf{x}) = p(x_1, x_2, \ldots, x_d) = p_1(x_1)p_2(x_2)\cdots p_d(x_d)$. This distribution is represented by a product node with $d$ children, the $i$th of which is a univariate distribution over $x_i$. Initially, we assume that the variables are independent, and the algorithm will update this probability distribution as new data points are processed.

Given a mini-batch of data points, the algorithm passes the points through the network from the root to the leaf nodes and updates each node along the way. This update includes two parts: i) updating the parameters of the SPN, and ii) updating the structure of the network.

## 3.1   Parameter update

There are two types of parameters in the model: weights on the branches under a sum node, and parameters for the Gaussian distribution in a leaf node. We use an online version of the hard EM algorithm to update the network parameters [13]. We prove that the algorithm monotonically improves the likelihood of the last data point. We also extend it to work for Gaussian leaf nodes. The pseudocode of this procedure (Alg. 1) is provided in the supplementary material.

Every node in the network has a count, $n_c$, initialized to 1. When a data point is received, the likelihood of this data point is computed at each node. Then the parameters of the network are updated in a recursive top-down fashion by starting at the root node. When a sum node is traversed, its count is increased by 1 and the count of the child with the highest likelihood is increased by 1. In a feed-forward network, the weight $w_{s,c}$ of a branch between a sum node $s$ and one of its children $c$ is estimated as $w_{s,c} = \frac{n_c}{n_s}$ where $n_s$ is the count of the sum node and $n_c$ is the count of the child node. We recursively update the subtree of the child with the highest likelihood.

We recursively update the subtrees rooted at each child of a product node. For Gaussian leaf nodes, we keep track of the empirical mean vector $\mu$ and covariance matrix $\Sigma$ for the variables in their scope. When a leaf node with a current count of $n$ receives a batch of $m$ data points $x^{(1)}, x^{(2)}, \ldots, x^{(m)}$, the empirical mean $\mu$ and covariance $\Sigma$ are updated according to the following equations:

$$\mu_i' = \frac{1}{n+m}\left(n\mu_i + \sum_{k=1}^{m} x_i^{(k)}\right) \tag{1}$$

$$\Sigma_{i,j}' = \frac{1}{n+m}\left[n\Sigma_{i,j} + \sum_{k=1}^{m}\left(x_i^{(k)} - \mu_i\right)\left(x_j^{(k)} - \mu_j\right)\right] - (\mu_i' - \mu_i)(\mu_j' - \mu_j)$$

where $i$ and $j$ index the variables in the leaf node's scope.

The update of these sufficient statistics can be seen as locally maximizing the likelihood of the data. The empirical mean and covariance of the Gaussian leaves locally increase the likelihood of the data that reach that leaf. Similarly, the count ratios used to set the weights under a sum node locally increase the likelihood of the data that reach each child. We prove this result below.

**Theorem 1.** *Let $\theta_s$ be the set of parameters of an SPN $s$, and let $f_s(\cdot|\theta_s)$ be the probability density function of the SPN. Given an observation $x$, suppose the parameters are updated to $\theta_s'$ based on the running average update procedure, then $f_s(x|\theta_s') \geq f_s(x|\theta_s)$.*

*Proof.* We will prove the theorem by induction. First suppose the SPN is just one leaf node. In this case, the parameters are the empirical mean and covariance, which is the maximum likelihood estimator for a Gaussian distribution. Suppose $\theta$ consists of the parameters learned using $n$ data points $x^{(1)}, \ldots, x^{(n)}$, and $\theta'$ consists of the parameters learned using the same $n$ data points and an additional observation $x$. Then we have

$$f_s(x|\theta'_s) \prod_{i=1}^n f_s(x^{(i)}|\theta'_s) \geq f_s(x|\theta_s) \prod_{i=1}^n f_s(x^{(i)}|\theta_s) \geq f_s(x|\theta_s) \prod_{i=1}^n f_s(x^{(i)}|\theta'_s) \qquad (2)$$

Thus we get $f_s(x|\theta'_s) \geq f_s(x|\theta_s)$. Suppose we have an SPN $s$ where each child SPN $t$ satisfies the property $f_t(x|\theta'_t) \geq f_t(x|\theta_t)$. If the root of $s$ is a product node, then $f_s(x|\theta'_s) = \prod_t f_t(x|\theta'_t) \geq \prod_t f_t(x|\theta_t) = f_s(x|\theta_s)$. Now suppose the root of $s$ is a sum node. Let $n_t$ be the count of child $t$, and let $u = \arg\max_t f_t(x|\theta_t)$ be the child with the highest count. Then we have

$$f_s(x|\theta'_s) = \frac{1}{n+1} \left( f_u(x|\theta'_u) + \sum_t n_t f_t(x|\theta'_t) \right) \geq \frac{1}{n+1} \left( f_u(x|\theta_u) + \sum_t n_t f_t(x|\theta_t) \right)$$

$$\geq \frac{1}{n+1} \left( \sum_t \frac{n_t}{n} f_t(x|\theta_t) + \sum_t n_t f_t(x|\theta_t) \right) = \frac{1}{n} \sum_t n_t f_t(x|\theta_t) = f_s(x|\theta_s) \quad \square$$

### 3.2   Structure update

The simple online parameter learning technique described above can be easily extended to enable online structure learning. In the supplementary material, Alg. 2 describes the pseudocode of the resulting procedure called oSLRAU (online Structure Learning with Running Average Update). Similar to leaf nodes, each product node also keeps track of the empirical mean vector and empirical covariance matrix of the variables in its scope. These are updated in the same way as the leaf nodes.

Initially, when a product node is created using traditional structure learning, all variables in the scope are assumed independent (see Alg. 3 in the supplementary material). As new data points arrive at a product node, the covariance matrix is updated, and if the absolute value of the Pearson correlation coefficient between two variables are above a certain threshold, the algorithm updates the structure so that the two variables become correlated in the model.

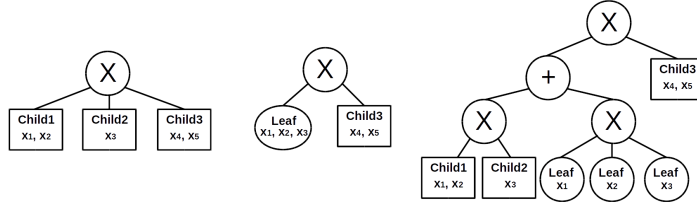

Figure 1: Depiction of how correlations between variables are introduced. Left: original product node with three children. Middle: combine Child1 and Child2 into a multivariate leaf node (Alg. 4). Right: create a mixture to model the correlation (Alg. 5).

We correlate two variables in the model by combining the child nodes whose scopes contain the two variables. The algorithm employs two approaches to combine the two child nodes: a) create a multivariate leaf node (Alg. 4 in the supplementary material), or b) create a mixture of two components over the variables (Alg. 5 in the supplementary material). These two processes are depicted in Figure 1. On the left, a product node with scope $x_1, \ldots, x_5$ originally has three children. The product node keeps track of the empirical mean and covariance for these five variables. Suppose it receives a mini-batch of data and updates the statistics. As a result of this update, $x_1$ and $x_3$ now have a correlation above the threshold. In the middle of Figure 1, the algorithm combines the two child nodes that have $x_1$ and $x_3$ in their scope, and turns them into a multivariate leaf node. Since the product node already keeps track of the mean and covariance of these variables, we can simply use those statistics as the parameters for the new leaf node.

Another way to correlate $x_1$ and $x_3$ is to create a mixture, as shown in Figure 1(right). The mixture has two components. The first contains the original children of the product node that contain $x_1$ and $x_3$. The second component is a new product node, which is again initialized to have a fully factorized

distribution over its scope (see Alg. 3 in the supplementary material). The mini-batch of data points are then passed down the new mixture to update its parameters. Although the children are drawn like leaf nodes in the diagrams, they can in fact be entire subtrees. Since the process does not involve the parameters of a child, it works the same way if some of the children are trees instead of single nodes.

The technique chosen to induce a correlation depends on the number of variables in the scope. The algorithm creates a multivariate leaf node when the combined scope of the two children has a number of variables that does not exceed some threshold and if the total number of variables in the problem is greater than this threshold, otherwise it creates a mixture. Since the number of parameters in multivariate Gaussian leaves grows at a quadratic rate with respect to the number of variables, it is not advised to consider multivariate leaves with too many variables. In contrast, the mixture construction increases the number of parameters at a linear rate.

To simplify the structure, if a product node ends up with only one child, it is removed from the network, and its only child is joined with its parent. Similarly, if a sum node ends up being a child of another sum node, then the child sum node can be removed, and all its children are promoted one layer up. We also prune subtrees periodically when the count at a node does not increase for several mini-batches. This helps to prevent overfitting and to adapt to changes in non-stationary settings.

Note that this structure learning technique does a single pass through the data and therefore is entirely online. The time and space complexity of updating the structure after each data point is linear in the size of the network (i.e., # of edges) and quadratic in the number of features (since product nodes store a covariance matrix that is quadratic in the size of their scope). The algorithm also ensures that the decomposability and completeness properties are preserved after each update.

### 3.3  Updates in Recurrent Networks

We can generalize the parameter and structure updates described in the previous sections to handle recurrent SPNs as follows. We start with a bottom network that has $k$ fully factored distributions. The template network initially has $k$ interface input product nodes, an intermediate layer of $k$ sum nodes and an output interface layer of $k$ product nodes. Fig. 2(top) shows an initial template network when $k = 2$. The top network consists of a single sum node linked to the output interface layer of the template network. For the parameter updates, we unroll the recurrent SPN by creating as many copies of the template network as needed to match the length of a data sequence. Fig. 2(bottom) shows an unrolled recurrent SPN over 3 time steps. We use a single shared count for each node of the template network even though template nodes are replicated multiple times. A shared count is incremented each time a data sequence goes through its associated node in any copy of the template network. Similarly, the empirical mean and covariance of each leaf in the template network are shared across all copies of the template network.

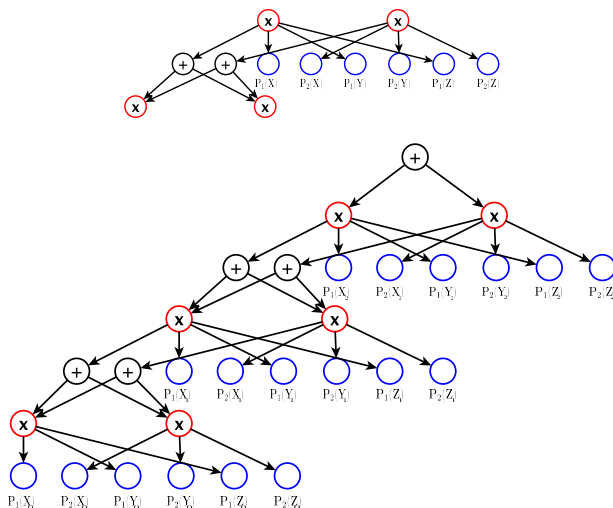

Figure 2: Top: A generic template network with interface nodes drawn in red and leaf distributions drawn in blue. Bottom: A recurrent SPN unrolled over 3 time steps.

Structure updates in recurrent networks can also be done by detecting correlations between pairs of variables that are not already captured by the network. A single shared covariance matrix is estimated at each product node of the template network. To circumvent the fact that the scope of a product node will differ in each copy of the template network, we relabel the scope of each input interface node to a single unique binary latent variable that takes value 1 when a data sequence traverses this node and 0 otherwise. These latent binary variables can be thought of as summarizing the information below the input interface nodes of each copy of the template network. This ensures that the variables in each copy of the template network are equivalent and therefore we can maintain a shared covariance matrix at each product node of the template network. When a significant correlation is detected between the variables in the scope of two different children of a product node, a mixture is introduced as depicted in the right part of Fig. 1.

## 4 Experiments

We compare the performance of oSLRAU with other methods on both simple and larger data sets with continuous variables. We begin this section by describing the data sets.

### 4.1 Synthetic Data

As a proof of concept, we test the algorithm on a synthetic dataset. We generate data from a 3-dimensional distribution

$$p(x_1, x_2, x_3) = [0.25N(x_1|1, 1)N(x_2|2, 2) + 0.25N(x_1|11, 1)N(x_2|12, 2)$$
$$+ 0.25N(x_1|21, 1)N(x_2|22, 2) + 0.25N(x_1|31, 1)N(x_2|32, 2)]N(x_3|3, 3)$$

where $N(\cdot|\mu, \sigma^2)$ is the normal distribution with mean $\mu$ and variance $\sigma^2$. Therefore, the first two dimensions $x_1$ and $x_2$ are generated from a Gaussian mixture with four components, and $x_3$ is independent of the other two variables.

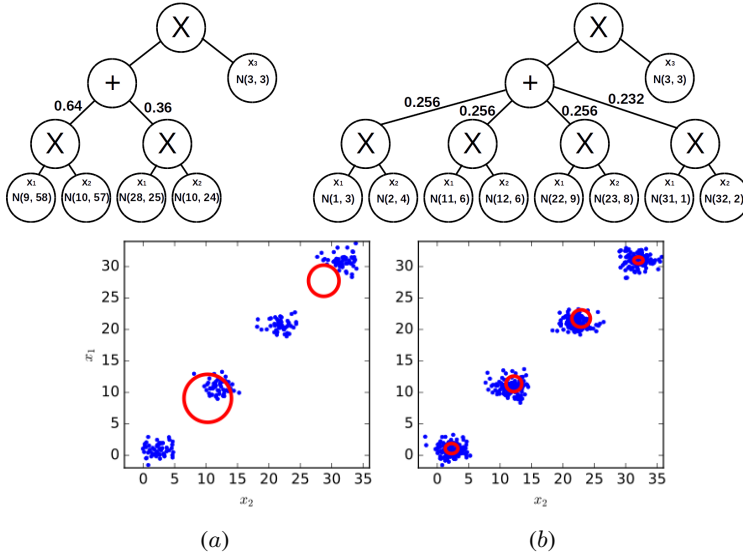

(a)                                      (b)

Figure 3: Learning the structure from the toy dataset using univariate leaf nodes after (a) 200 data points and (b) 500 data points. Blue dots are the data points from the toy dataset, and the red ellipses show diagonal Gaussian components learned.

Starting from a fully factorized distribution, we would expect $x_3$ to remain factorized after learning from data. Furthermore, the algorithm should generate new components along the first two dimensions as more data points are received since $x_1$ and $x_2$ are correlated. This is indeed observed in Figures 3a and 3b, which show the structure learned after 200 and 500 data points. The variable $x_3$ remains factorized regardless of the number of data points seen, whereas more components are created for $x_1$ and $x_2$ as more data points are processed. Bottom charts in Figures 3a and 3b show the data points along the first two dimensions and the Gaussian components learned. We observe that the algorithm generates new components to model the correlation between $x_1$ and $x_2$ as it processes more data.

## 4.2 Large Continuous Datasets

We also tested oSLRAU's combined parameter and structure updates on large real-world datasets with continuous features (see supplementary material for details about each dataset). Table 1 compares the average log-likelihood of oSLRAU to that of randomly generated networks and a modified version of ILSPN [9] that we adapted to Gaussian SPNs. For a fair comparison we generated random networks that are at least as large as the networks obtained by oSLRAU. Observe that oSLRAU achieves higher log-likelihood than random networks since it effectively discovers empirical correlations and generates a structure that captures those correlations. ILSPN ran out of memory for 3 problems where it generated networks of more than 7.5 Gb. It underperformed oSLRAU on two other problems since it never modifies its product nodes after creation and its online clustering technique is not suitable for streaming data as it requires fairly large batch sizes to create new clusters.

Table 1: Average log-likelihood scores with standard error on large real-world data sets. The best results among the online techniques (random, ILSPN, oSLRAU and RealNVP online) are highlighted in bold. Results for RealNVP offline are also included for comparison purposes. "−" indicates that ILSPN exceeded the memory limit of 7.5 Gb.

| Datasets | Random | ILSPN | oSLRAU | RealNVP Online | RealNVP Offline |
|---|---|---|---|---|---|
| Voxforge | -33.9 ± 0.3 | —- | **-29.6** ± 0.0 | -169.0 ± 0.6 | -168.2 ± 0.8 |
| Power | -2.83 ± 0.13 | **-1.85 ± 0.02** | -2.46 ± 0.11 | -18.70 ± 0.19 | -17.85 ± 0.22 |
| Network | -5.34 ± 0.03 | -4.71 ± 0.16 | **-4.27** ± 0.04 | -10.80 ± 0.02 | -7.89 ± 0.05 |
| GasSen | -114 ± 2 | —- | **-102** ± 4 | -748 ± 99 | -443 ± 64 |
| MSD | -538.8 ± 0.7 | —- | -531.4 ± 0.3 | **-362.4** ± 0.4 | -257.1 ± 2.03 |
| GasSenH | -21.5 ± 1.3 | -182.3 ± 4.5 | **-15.6** ± 1.2 | -44.5 ± 0.1 | 44.2 ± 0.1 |

Table 2: Large datasets: comparison of oSLRAU with and without periodic pruning.

| Dataset | log-likelihood | | time (sec) | | SPN size (# nodes) | |
|---|---|---|---|---|---|---|
| | no pruning | pruning | no pruning | pruning | no pruning | pruning |
| Power | -2.46 ± 0.11 | **-2.40 ± 0.18** | 183 | 39 | 23360 | 5330 |
| Network | -4.27 ± 0.02 | **-4.20 ± 0.09** | 14 | 12 | 7214 | 5739 |
| GasSen | **-102 ± 4** | -130 ± 3 | 351 | 276 | 5057 | 1749 |
| MSD | -527.7 ± 0.28 | **-526.8 ± 0.27** | 74 | 72 | 1442 | 1395 |
| GasSenH | **-15.6 ± 1.2** | -17.7 ± 1.58 | 12 | 10 | 920 | 467 |

We also compare oSLRAU to a publicly available implementation of RealNVPwhich is a different type of generative neural network used for density estimation [5]. Since the benchmarks include a variety of problems from different domains and it is not clear which network architecture would work best, we used a default 2-hidden-layer fully connected network. The two layers have the same size. For a fair comparison, we used a number of nodes per layer that yields approximately the same number of parameters as the SPNs. Training was done by stochastic gradient descent in TensorFlow with a step size of 0.01 and mini-batch sizes that vary from 100 to 1500 depending on the size of the dataset. We report the results for online learning (single iteration) and offline learning (when validation loss stops decreasing). In this experiment, the correlation threshold was kept constant at 0.1. To determine the maximum number of variables in multivariate leaves, we utilized the following rule: at most one variable per leaf if the problem has 3 features or less and then increase the maximum number of variables per leaf up to 4 depending on the number of features. Further analysis on the effects of varying the maximum number of variables per leaf is included in the supplementary material. oSLRAU outperformed RealNVP on 5 of the 6 datasets. This can be explained by the fact that oSLRAU learns a structure that is suited for each problem while RealNVP does not learn any structure. Note that RealNVP may yield better results by using a different architecture than the default of 2-hidden layers, however in the absence of domain knowledge this is difficult. Furthermore, online learning with streaming data precludes an offline search over some hyperparameters such as the number of layers and nodes in order to refine the architecture. Hence, the results presented in Table 1 highlight the importance of an online learning technique such as oSLRAU to obtain a suitable network structure with streaming data in the absence of domain knowledge.

Table 2 reports the training time (seconds) and the size (# of nodes) of the SPNs constructed for each dataset by oSLRAU with and without periodic pruning. After every 1% of a dataset is processed, subtrees that have not been updated in the last percent of the dataset are pruned. This helps to mitigate overfitting while decreasing the size of the SPNs. The experiments were carried out on an Amazon

c4.xlarge machine with 4 vCPUs (high frequency Intel Xeon E5-2666 v3 Haswell processors) and 7.5 Gb of RAM. The times are short since oSLRAU does a single pass through the data.

Additional experiments are included in the supplementary material to evaluate the effect of the hyperparameters. Additional empirical comparisons between oSLRAU and other techniques are also presented in the supplementary material.

### 4.3 Nonstationary Generative Learning

We evaluate the effectiveness of the periodic pruning technique to adapt to changes in a nonstationary environment by feeding oSLRAU with a stream of 50,000 images from the MNIST dataset ordered by their label from 0 to 9. The bottom row of Fig. 4 shows a sample of images generated by the SPN (14,000 nodes) constructed by oSLRAU with pruning after every 6000 images. As the last images in the stream are 8 and 9, oSLRAU pruned parts of its network related to other digits and it generated mostly 9's. When pruning is disabled, the top row of Fig. 4 shows that the SPN (17,000 nodes) constructed by oSLRAU can generate a mixture of digits as it learned to generate all digits.



Figure 4: Top row: sample images generated by SPN learned by oSLRAU without pruning. Bottom: sample images generated by SPN learned by oSLRAU with pruning every 6000 images.

### 4.4 Sequence Data

We also tested oSLRAU's ability to learn the structure of recurrent SPNs. Table 3 reports the average log likelihood based on 10-fold cross validation with 5 sequence datasets. The number of sequences, the average length of the sequences and the number of observed variables is reported under the name of each dataset. We compare oSLRAU to the previous search-and-score (S&S) technique [10] for recurrent SPNs (RSPNs) with Gaussian leaves as well as HMMs with mixture of Gaussians emission distributions and recurrent neural networks (RNNs) with LSTM units and output units that compute the mean of Gaussians. The number of interface nodes in RSPNs, hidden states in HMMs and LSTM units in RNNs was bounded to 15. Parameter and structure learning was performed for the RSPNs while only parameter learning was performed for the HMMs and RNNs. The RNNs were trained by minimizing squared loss, which is mathematically equivalent to maximizing the data likelihood when we interpret each output as the mean of a univariate Gaussian. The variance of each Gaussian was optimized by a grid search in [0.01,0.1] in increments of 0.01 and in [0.1,2] in increments of 0.1. We did this solely for the purpose of reporting the log likelihood of the test data with RNNs, which would not be possible otherwise. oSLRAU outperformed the other techniques on 4 of the 5 datasets. It learned better structures than S&S in less time. oSLRAU took less than 11 minutes per dataset while S&S took 1 day per dataset.

Table 3: Average log-likelihood and standard error based on 10-fold cross validation. (#i,length,#oVars) indicates the number of data instances, average length of the sequences and number of observed variables per time step.

| Dataset (#i,length,#oVars) | hillValley (600,100,1) | eegEye (14970,14,1) | libras (350,90,1) | JapanVowels (270,16,12) | ozLevel (2170,24,2) |
|---|---|---|---|---|---|
| HMM | 286 ± 6.9 | 22.9 ± 1.8 | -116.5 ± 2.2 | -275 ± 13 | -34.6 ± 0.3 |
| RNN | 205 ± 23 | 15.2 ± 3.9 | -92.9 ± 12.9 | -257 ± 35 | **-15.3** ± 0.8 |
| RSPN+S&S | 296 ± 16.1 | 25.9 ± 2.1 | -93.5 ± 7.2 | -241 ± 12 | -34.4 ± 0.4 |
| RSPN+oSLRAU | **299.5** ± 18 | **36.9** ± 1.4 | **-83.5** ± 5.4 | **-231** ± 12 | -30.1 ± 0.4 |

## 5 Conclusion and Future work

This paper describes a new *online* structure learning technique for feed-forward and recurrent SPNs. oSLRAU can learn the structure of SPNs in domains for which it is unclear what might be a good structure, including sequence datasets of varying length. This algorithm can also scale to large datasets efficiently. We plan to extend this work by learning the structure of SPNs in an *online* and *discriminative* fashion. Discriminative learning is essential to attain good accuracy in classification.

## Acknowledgments

This research was funded by Huawei Technologies and NSERC. Prashant Doshi acknowledges support from NSF grant #IIS-1815598.

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
