[Supplementary Material]

# Online Structure Learning for Feed-Forward and Recurrent Sum-Product Networks: Supplementary Material

**Agastya Kalra,**[*]  **Abdullah Rashwan,**[*]  **Wilson Hsu,  Pascal Poupart**
Cheriton School of Computer Science, Waterloo AI Institute, University of Waterloo, Canada
Vector Institute, Toronto, Canada
agastya.kalra@gmail.com,{arashwan,wwhsu,ppoupart}@uwaterloo.ca

**Prashant Doshi**
Department of Computer Science
University of Georgia, USA
pdoshi@cs.uga.edu

**George Trimponias**
Huawei Noah's Ark Lab, Hong Kong
g.trimponias@huawei.com

## 1   Pseudocode

We include the pseudocode of the algorithms described in the paper:

- Algorithm 1: parameter update
- Algorithm 2: oSLRAU
- Algorithm 3: creating a factored distribution
- Algorithm 4: creating a multivariate Gaussian
- Algorithm 5: creating a mixture

---

**Algorithm 1** parameterUpdate(root(SPN),data)

---

**Input:** SPN and $m$ data points
**Output:** SPN with updated parameters
$\quad n_{root} \leftarrow n_{root} + m$
$\quad$**if** $isProduct(root)$ **then**
$\quad\quad$**for** each $child$ of $root$ **do**
$\quad\quad\quad parameterUpdate(child, data)$
$\quad\quad$**end for**
$\quad$**else if** $isSum(root)$ **then**
$\quad\quad$**for** each $child$ of $root$ **do**
$\quad\quad\quad subset \leftarrow \{x \in data \mid likelihood(child, x) \geq likelihood(child', x) \; \forall child' \text{ of } root\}$
$\quad\quad\quad parameterUpdate(child, subset)$
$\quad\quad\quad w_{root,child} \leftarrow \frac{n_{child}}{n_{root}}$
$\quad\quad$**end for**
$\quad$**else if** $isLeaf(root)$ **then**
$\quad\quad$update mean $\mu^{(root)}$ based on Eq. 2
$\quad\quad$update covariance matrix $\Sigma^{(root)}$ based on Eq. 3
$\quad$**end if**

---

[*]Equal contribution, first author was selected based on a coin flip

**Algorithm 2** $oSLRAU(root(SPN), data)$

---

**Input:** SPN and $m$ data points
**Output:** SPN with updated parameters

  $n_{root} \leftarrow n_{root} + m$
  **if** $isProduct(root)$ **then**
    update covariance matrix $\Sigma^{(root)}$
    $highestCorrelation \leftarrow 0$
    **for** each $c, c' \in children(root)$ where $c \neq c'$ **do**

$$correlation_{c,c'} \leftarrow \max_{i \in scope(c), j \in scope(c')} \frac{|\Sigma_{ij}^{(root)}|}{\sqrt{\Sigma_{ii}^{(root)} \Sigma_{jj}^{(root)}}}$$

      **if** $correlation_{c,c'} > highestCorrelation$ **then**
        $highestCorrelation \leftarrow correlation_{c,c'}$
        $child_1 \leftarrow c$
        $child_2 \leftarrow c'$
      **end if**
    **end for**
    **if** $highest \geq threshold$ **then**
      **if** $|scope(child_1) \cup scope(child_2)| \geq nVars$ **then**
        $createMixture(root, child_1, child_2)$
      **else**
        $createMultivariateGaussian(root, child_1, child_2)$
      **end if**
    **end if**
    **for** each $child$ of $root$ **do**
      $oSLRAU(child, data)$
    **end for**
  **else if** $isSum(root)$ **then**
    **for** each $child$ of $root$ **do**
      $subset \leftarrow \{x \in data \mid likelihood(child, x) \geq likelihood(child', x) \; \forall child' \text{ of } root\}$
      $oSLRAU(child, subset)$
      $w_{root,child} \leftarrow \frac{n_{child}+1}{n_{root}+\#children}$
    **end for**
    **if** #points seen modulo f equals 0 **then**
      **for** each $child$ of $root$ **do**
        **if** $n_{child} \leq 1$ **then**
          $RemoveChild(child)$
        **end if**
      **end for**
      $n_{root} \leftarrow 1$
      **for** each $child$ of $root$ **do**
        $n_{child} \leftarrow \frac{n_{child}+1}{n_{root}+\#children}$
      **end for**
    **end if**
  **else if** $isLeaf(root)$ **then**
    update mean $\mu^{(root)}$
    update covariance matrix $\Sigma^{(root)}$
  **end if**

---

---
**Algorithm 3** $createFactoredModel(scope)$

---
**Input:** scope (set of variables)
**Output:** fully factored SPN
  $factoredModel \leftarrow$ create product node
  **for** each $i \in scope$ **do**
    add $N_i(\mu{=}0, \sigma{=}\Sigma_{i,i}^{(root)})$ as child of $factoredModel$
  **end for**
  $\Sigma^{(factoredModel)} \leftarrow \mathbf{0}$
  $n_{factoredModel} \leftarrow 0$
  return $factoredModel$

---

---
**Algorithm 4** $createMultiVarGaussian(root, child_1, child_2)$

---
**Input:** SPN, two children to be merged and data
**Output:** new multivariate Gaussian
  create $multiVarGaussian$
  $jointScope \leftarrow \{scope(child_1) \cup scope(child_2)\}$
  $\mu^{(multiVarGaussian)} \leftarrow \mu_{jointScope}^{(root)}$
  $\Sigma^{(multiVarGaussian)} \leftarrow \Sigma_{jointScope,jointScope}^{(root)}$
  $n_{multiVarGaussian} \leftarrow n_{root}$
  return $multiVarGaussian$

---

---
**Algorithm 5** $createMixture(root, child_1, child_2)$

---
**Input:** SPN and two children to be merged
**Output:** new mixture model
  remove $child_1$ and $child_2$ from $root$
  $component_1 \leftarrow$ create product node
  add $child_1$ and $child_2$ as children of $component_1$
  $n_{component_1} \leftarrow n_{root}$
  $jointScope \leftarrow scope(child_1) \cup scope(child_2)$
  $\Sigma^{(component_1)} \leftarrow \Sigma_{jointScope,jointScope}^{(root)}$
  $component_2 \leftarrow createFactoredModel(jointScope)$
  $n_{component_2} \leftarrow 0$
  $mixture \leftarrow$ create sum node
  add $component_1$ and $component_2$ as children of $mixture$
  $n_{mixture} \leftarrow n_{root}$
  $w_{mixture,component_1} \leftarrow \frac{n_{component_1}+1}{n_{mixture}+2}$
  $w_{mixture,component_2} \leftarrow \frac{n_{component_2}+1}{n_{mixture}+2}$
  add $mixture$ as child of $root$
  return $root$

---

## 2 Parameter Learning technique

Alg. 1 does a single pass through the data. The complexity of updating the parameters after each data point is linear in the size of the network (i.e., # of edges) since it takes one bottom up pass to compute the likelihood of the data point at each node and one top-down pass to update the sufficient statistics and the weights.

## 3 Experiments

### 3.1 Size of Datasets

Table 1: Information for each large dataset

| Dataset | Datapoints | Variables |
|---------|------------|-----------|
| Voxforge | 3,603,643 | 39 |
| Power | 2,049,280 | 4 |
| Network | 434,873 | 3 |
| GasSen | 8,386,765 | 16 |
| MSD | 515,344 | 90 |
| GasSenH | 928,991 | 10 |

### 3.2 Comparison to other Algorithms

In a second experiment, we compare our algorithm to several alternatives on the same datasets used by [2]. We use 0.1 as the correlation threshold in all experiments, and we use mini-batch sizes of 1 for the three datasets with fewest instances (Quake, Banknote, Abalone), 8 for the two slightly larger ones (Kinematics, CA), and 256 for the two datasets with most instances (Flow Size, Sensorless).

The experimental results for our algorithm called *online structure learning with running average update* (oSLRAU) are listed in Table **??** along with results reproduced from [2]. The table reports the average test log likelihoods with standard error on 10-fold cross validation. oSLRAU achieved better log likelihoods than online Bayesian moment matching (oBMM) [2] and online expectation maximization (oEM) [1] with network structures generated at random or corresponding to Gaussian mixture models (GMMs). This highlights the main advantage of oSLRAU: learning a structure that models the data. Stacked Restricted Boltzmann Machines (SRBMs) [4] and Generative Moment Matching Networks (GenMMNs) [3] are other types of deep generative models. Since it is not possible to compute the likelihood of data points with GenMMNs, the model is augmented with Parzen windows. More specifically, 10,000 samples are generated using the resulting GenMMNs and a Gaussian kernel is estimated for each sample by adjusting its parameters to maximize the likelihood of a validation set. However, as pointed out by [5] this method only provides an approximate estimate of the log-likelihood and therefore the log-likelihood reported for GenMMNs in Table **??** may not be directly comparable to the log-likelihood of other models.

The network structures for GenMMNs and SRBMs are fully connected while ensuring that the number of parameters is comparable to those of the SPNs. oSLRAU outperforms these models on 5 datasets while SRBMs and GenMMNs each outperform oSLRAU on one dataset. Although SRBMs and GenMMNs are more expressive than SPNs since they allow other types of nodes beyond sums and products, training GenMMNs and SRBMs is notoriously difficult. In contrast, oSLRAU provides a simple and effective way of optimizing the structure and parameters of SPNs that captures well the correlations between variables and therefore yields good results.

### 3.3 Hyperparameter Search

To understand the impact that the maximum number of variables per leaf node has on the resulting SPN, we performed experiments where the minibatch size and correlation threshold were held constant for a given dataset while the maximum number of variables per leaf node varies. We report the log likelihood with standard error after ten-fold cross validation, as well as average size and average time in Tables 3, 4 and 5. As expected, the number of nodes in an SPN decreases as the leaf node cap increases, since there will be less branching. What's interesting is that depending on the type of correlations in the datasets, different sizes perform better or worse. For example in Power, we notice that univariate leaf nodes are the best, but in GasSenH, slightly larger leaf nodes tend to do well. We show that too many variables in a leaf node leads to worse performance and underfitting, and in some cases too few variables per leaf node leads to overfitting. These results show that in general, the largest decrease in size and time while maintaining good performance occurs with a maximum of 3 variables per leaf node. Therefore in practice, 3 variables per leaf node works well, except when there are only a few variables in the dataset, then 1 is a good choice.

Table 3: Log likelihoods with standard error as we vary the threshold for the maximum # of variables in a multivariate Gaussian leaf. No results are reported (dashes) when the maximum # of variables is greater than the total number of variables.

| Dataset | Maximum # of Variables per Leaf Node | | | | |
|---|---|---|---|---|---|
| | 1 | 2 | 3 | 4 | 5 |
| Power | **-1.71** $\pm$ 0.18 | -3.02 $\pm$ 0.24 | -3.74 $\pm$ 0.28 | -4.52 $\pm$ 0.1 | —— |
| Network | **-4.27** $\pm$ 0.09 | -4.53 $\pm$ 0.09 | -4.75 $\pm$ 0.02 | —— | —— |
| GasSen | -105 $\pm$ 2.5 | -103 $\pm$ 2.8 | **-102** $\pm$ 4.1 | -104 $\pm$ 3.8 | -103 $\pm$ 3.8 |
| MSD | -532 $\pm$ 0.32 | **-531** $\pm$ 0.32 | **-531** $\pm$ 0.28 | **-531** $\pm$ 0.31 | -532 $\pm$ 0.34 |
| GasSenH | -17.2 $\pm$ 1.04 | -16.8 $\pm$ 1.23 | **-15.6** $\pm$ **1.13** | -15.9 $\pm$ 1.3 | -16.1 $\pm$ 1.47 |

Table 4: Average times (seconds) as we vary the threshold for the maximum # of variables in a multivariate Gaussian leaf. No results are reported (dashes) when the maximum # of variables is greater than the total number of variables.

| Dataset | Maximum # of Variables per Leaf Node | | | | |
|---|---|---|---|---|---|
| | 1 | 2 | 3 | 4 | 5 |
| Power | 133 | 41.5 | 13.8 | 9.9 | —— |
| Network | 14.1 | 4.01 | 1.92 | —— | —— |
| GasSen | 783.78 | 450.34 | 350.52 | 148.89 | 145.759 |
| MSD | 80.47 | 64.44 | 44.9 | 43.65 | 41.44 |
| GasSenH | 16.59 | 13.35 | 11.76 | 11.04 | 10.16 |

Tables 6, 7 and 8 show respectively how the log-likelihood, time and size changes as we vary the correlation threshold from 0.05 to 0.7. A very small correlation threshold tends to detect spurious correlations and lead to overfitting while a large correlation threshold tends to miss some correlations and lead to underfitting. The results in Table 6 generally support this tendency subject to noise due to sample effects. Since the highest log-likelihood was achieved in three of the datasets with a correlation threshold of 0.1, this explains why we used 0.1 as the threshold in the previous experiments. Tables 7 and 8 also show that the average time and size of the resulting SPNs generally decrease (subject to noise) as the correlation threshold increases since fewer correlations tend to be detected.

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

Table 5: Average SPN sizes (# of nodes) as we vary the threshold for the maximum # of variables in a multivariate Gaussian leaf. No results are reported (dashes) when the maximum # of variables is greater than the total number of variables.

| Dataset | Maximum # of Variables per Leaf Node | | | | |
|---|---|---|---|---|---|
| | 1 | 2 | 3 | 4 | 5 |
| Power | 14269 | 2813 | 427 | 8 | —— |
| Network | 7214 | 1033 | 7 | —— | —— |
| GasSen | 13874 | 6879 | 5057 | 772 | 738 |
| MSD | 6547 | 3114 | 802 | 672 | 582 |
| GasSenH | 1901 | 1203 | 920 | 798 | 664 |

Table 6: Log Likelihoods for different correlation thresholds.

| Dataset | Correlation Threshold | | | | | |
|---|---|---|---|---|---|---|
| | 0.05 | 0.1 | 0.2 | 0.3 | 0.5 | 0.7 |
| Power | -2.37 ± 0.13 | -2.46 ± 0.11 | **-2.20** ± 0.18 | -3.02 ± 0.24 | -4.65 ± 0.11 | -4.68 ± 0.09 |
| Network | **-3.98** ± 0.09 | -4.27 ± 0.02 | -4.75 ± 0.02 | -4.75 ± 0.02 | -4.75 ± 0.02 | -4.75 ± 0.02 |
| GasSen | -104 ± 5 | **-102** ± 4 | **-102** ± 3 | **-102** ± 3 | -103 ± 3 | -110 ± 3 |
| MSD | **-531.4** ± 0.3 | **-531.4** ± 0.3 | **-531.4** ± 0.3 | **-531.4** ± 0.3 | -532.0 ± 0.3 | -536.2 ± 0.1 |
| GasSenH | **-15.6** ± 1.2 | **-15.6** ± 1.2 | -15.8 ± 1.1 | -16.2 ± 1.4 | -16.1 ± 1.4 | -17.2 ± 1.4 |

Table 7: Average times (seconds) as we vary the correlation threshold.

| Dataset | Correlation Threshold | | | | | |
|---|---|---|---|---|---|---|
| | 0.05 | 0.1 | 0.2 | 0.3 | 0.5 | 0.7 |
| Power | 197 | 183 | 130 | 39 | 10 | 9 |
| Network | 20 | 14 | 1.9 | 1.9 | 1.9 | 1.9 |
| GasSen | 370 | 351 | 349 | 366 | 423 | 142 |
| MSD | 44.3 | 43.7 | 44.3 | 44.0 | 43.0 | 30.3 |
| GasSenH | 11.8 | 11.7 | 11.9 | 13.0 | 12.0 | 15.1 |

[4] Salakhutdinov, Ruslan and Hinton, Geoffrey E. Deep boltzmann machines. In *AISTATS*, pp. 448–455, 2009.

[5] Theis, Lucas, Oord, Aäron, and Bethge, Matthias. A note on the evaluation of generative models. *arXiv:1511.01844*, 2015.

Table 8: Average SPN sizes (# of nodes) as the correlation threshold changes.

| Dataset | Correlation Threshold | | | | | |
|---|---|---|---|---|---|---|
| | 0.05 | 0.1 | 0.2 | 0.3 | 0.5 | 0.7 |
| Power | 24914 | 23360 | 16006 | 2813 | 11 | 11 |
| Network | 11233 | 7214 | 9 | 9 | 9 | 9 |
| GasSen | 5315 | 5057 | 5041 | 5035 | 4581 | 490 |
| MSD | 672 | 672 | 674 | 674 | 660 | 448 |
| GasSenH | 920 | 920 | 887 | 877 | 1275 | 796 |