[Reviews · NeurIPS 2018]

Reviewer 1



Incremental approach to learn SPNs based on new observations by adapting the structure (using expansions and collapsing of nodes) and parameters accordingly. The main result is that one can do so while improving the model quality measure, which leads to an interesting updating procedure for SPNs (though one may debate about the term "online" for that). The approach to learn SPNs expands and collapses the model according to incoming observations. The idea is somehow ad-hoc, but it is interesting nevertheless. Experiments show good accuracy in a collection of cases. The drawback is that SPNs are arguably more suitable for discriminative instead of generative analysis. This is acknowledged in the conclusions. The paper is well-written and the approach is interesting. Learning SPNs in an "online" fashion is important and worth pursuing. Some points: - Abstract: "inference is always tractable" - this is not true, full MAP is already NP-hard. - "SPNs are equivalent to Bayesian and Markov networks" - this might be misleading, since to represent the same domain, one may need an exponentially large SPN in order to represent the same as a BN or MRF. So we should be careful with the "equivalence" usage. - Hard EM, which is mentioned for learning, is an NP-hard task in SPNs, since it requires (full) MAP inferences (see Peharz et al PAMI 2016, Conaty et al UAI 2017, etc). Obviously one cannot claim that an approach is tractable if part of the task is NP-hard and an approximate algorithm is used. It would be better to clarify that. - Some references in the list are incomplete. After response: The issue with respect to MAP and hard EM has been clarified.

Reviewer 2



The paper presents a new algorithm for online parameter and structure learning of SPNs. The algorithms are tested on a fairly large selection of different problems and situations, and compare favorably against other SPN learning algorithms and (in some scenarios) other deep models. The proposed technique is rather simple, but seems to be effective. The experiments in particular are convincing. The text flows nicely. It is unfortunate that Theo 1 which shows that the parameter learning update improves on loglikehood is not in the paper; this result should be at least stated in the text.

Reviewer 3



This paper proposed an online learning algorithm for static and dynamic sum-product networks (SPNs), a type of probabilistic model with tractable inference. The authors essentially combine local structure search in SPNs with a hard variant of expectation-maximization [1]. The algorithm maintains empirical covariance estimates of product nodes and leverages statistical dependence tests to decide when to replace a product (factorized distribution) with either a new leaf or a mixture (sum node). The algorithm further includes a pruning mechanism in order to trim over-grown structures. The proposed method is called online Structure Learning with Running Average Update (oSLRAU). Strengths: + the hard EM scheme, already used by Poon and Domingos [1], is an effective and efficient method for parameter learning in SPNs. While Poon and Domingos applied it to select a sub-structure in a pre-determined layout, its use for flexible structure learning is good idea. + the structure updates by monitoring empirical means and covariances at product nodes is also a natural and elegant choice. + together with network pruning, the proposed structure learning scheme is natural and has the potential to become a state-of-the-art method. Weaknesses: - while the approach has certain potential strengths, see above, the paper does not elaborate on them in a satisfying way. This approach could break new ground in learning SPNs, but the paper is presented in a "yet-another-SPN-learning-algorithm" fashion. - the paper is overloaded: online learning of both parameters and structure in (static) SPNs is already an important topic far from being closed. The paper, however, additionally discusses dynamic SPNs in a somewhat superficial way. In that way, the paper is not focused on a particular problem. - the experimental evaluation is also rather unfocused. It is not clear to me why the particular tasks and comparison methods where chosen. Quality: The approach in this paper is reasonably justified. In particular, it leverages the hard EM variant due to Poon & Domingos [1], which was successful in selecting a sub-structure in a large potential structure layout tailored to images. In this paper, this method is used in a completely dynamic way, which I find promising. The structural updates (replacing product nodes with multi-variate Gaussians or new sum nodes; pruning) are also natural and remind at structural search in SPNs [2,3]. However, these advantages are not elaborated in a satisfying manner. I would really like to see a serious attempt to establish a new state-of-the-art method for learning SPNs; no theoretical justification/motivation for oSLRAU is given (besides a Theorem in the supplementary, guaranteeing that the likelihood of the last single sample is guaranteed to be increased -- does this really matter?), nor is it empirically demonstrated that it improves state-of-the-art. In particular, an eager comparison with LearnSPN is left out. Additionally I don't see the benefit to also discuss oSLRAU for dynamic SPNs, in particular, as these are discussed in a quite superficial and incomprehensible way. Clarity: The paper is reasonably clear, but could be improved at some parts. In particular, the paper does not give a focused impression (what is the goal of the paper), many choices are not motivated (e.g. why compare e.g. with NVP but not with LearnSPN), many relevant parts are deferred to the supplementary. Originality: The paper is quite original, but it does not discuss prior art in a fully adequate and careful way. Two examples: 1) "SPNs were first introduced in [Poon and Domgos, 2011] as new type of deep representation." While it is certainly true that Poon and Domingos introduced new insights and ideas to SPNs, it is crucial to acknowledge the work of Darwiche, e.g. [4]. 2) "SPNs are equivalent to Bayesian networks and Markov networks [Zhao et al., 2015], ..." First it is not at all explained in what sense SPNs are supposed to be equivalent to Bayesian networks and Markov networks, and second, this is not what Zhao et al. have discussed. SPNs are certainly not equivalent to Markov network (in whatever sense), and also not to Bayesian networks. Zhao et al. have shown the SPNs can be cast into special Bayesian networks incorporating a layer of latent variables. Furthermore, it is clear that Bayesian networks over discrete variables using CPTs can also be easily mapped to SPNs, but, clearly, somewhat more exotic Bayesian networks are not easily mapped to SPNs. Significance: The ideas in this paper have the potential to establish new state-of-the-art and are therefore of potential significance. However, as mentioned above, due to the rather unfocused presentation, motivation, and empirical evaluation, the paper does not unfold this potential. [1] Poon, H. and Domingos, P., "Sum-Product Networks: A new deep architecture", UAI, 2011. [2] Dennis, A. and Ventura, D., "Greedy Structure Search for Sum-Product Networks", IJCAI, 2015. [3] Peharz, R. and Gens, R. and Domingos, P. "Learning selective sum-product networks", LTPM, 2014. [4] Darwiche, A., "A differential approach to inference in Bayesian networks", ACM, 2003. ***EDIT*** The authors (and reviewer #1) have discussed NP-hardness of hard EM. Please note that hard EM is *not* NP-hard in SPNs. In fact, it is well tractable. To see this, assume that X are the observed variables and Z are the latent variables associated with the sum nodes. Hard-EM is like soft-EM, but where the expectation over the complete log-likelihood w.r.t. the posterior P(Z | X) is replaced with the MAP solution of P(Z | X), i.e. argmax_theta E_{P(Z | X ; theta_old)} [log P(X,Z)] (soft EM) becomes argmax_theta log P( X, argmax_Z(P(X, Z ; theta_old)) ) (hard EM) where theta are all parameters (sum-weights, params of input distributions). The critical bit is computing Z* = argmax_Z(P(X, Z ; theta_old)), i.e. the most likely joint state of the latent variables (which determines an induced tree, see Zhao et al. [1]) given a sample for X. This inference query is well tractable. In fact, already Poon and Domingos [2] gave the correct algorithm for this (without proof): simply replace sum nodes with max nodes in the upwards pass and perform max-backtracking. That this algorithm is indeed correct was proved by Peharz et al. [3], Theorem 2: While reviewer #1 is right that MAP inference is in general NP-hard, inferring Z* is a notable exception, since the augmented SPN (the SPN which makes the latent variables explicit [3]) is *selective*. Note that the augmented SPN does not need to be constructed explicitly. It is clear that given Z* for each sample, updating the parameters is easy, in fact closed form. Also note that the same (exact) inference of Z* as required for hard EM was used by Vergari et al. [4], with the goal to infer a data representation. The authors also discussed a variant of hard EM used by Poon and Domingos [2], where sums are used in the upwards pass. It is rather clear that this is a kind of nested hard EM: perform MAP inference of the top sum layer, with the latent variables below marginalized, fix (condition on) this assignment for the top layer, and continue with MAP inference in the next layer, and so forth. It would be desirable that the authors incorporate this well-established theory in their paper, since hard EM is the central working horse of their method. A further point of importance is the connection between Bayesian networks, Markov networks and SPNs. Reviewer #1 stated: """ "SPNs are equivalent to Bayesian and Markov networks" - this might be misleading, since to represent the same domain, one may need an exponentially large SPN in order to represent the same as a BN or MRF. So we should be careful with the "equivalence" usage. """ I stated: """ 2) "SPNs are equivalent to Bayesian networks and Markov networks [Zhao et al., 2015], ..." First it is not at all explained in what sense SPNs are supposed to be equivalent to Bayesian networks and Markov networks, and second, this is not what Zhao et al. have discussed. SPNs are certainly not equivalent to Markov network (in whatever sense), and also not to Bayesian networks. Zhao et al. have shown the SPNs can be cast into special Bayesian networks incorporating a layer of latent variables. Furthermore, it is clear that Bayesian networks over discrete variables using CPTs can also be easily mapped to SPNs, but, clearly, somewhat more exotic Bayesian networks are not easily mapped to SPNs. """ The authors were evidently not alerted by the fact that two reviewers were independently puzzled by their claim. According to common understanding, Markov networks and Bayesian networks are *not* equivalent, see any common text book on graphical models [5]. So they cannot both be equivalent to SPNs. The notion of equivalence used by the authors, by which two models are equivalent whenever they can be convert into each other -- possibly under an exponential blowup -- is not meaningful. Sure, a fully connected BN can represent any distribution, so can a fully connected Markov network. This does *not* make these models equivalent. The authors stated that they are not aware of more exotic variants of BNs which cannot obviously be converted into an SPN. Take for example a BN over continuous variables, whose conditional distributions are Gamma distributions, parametrized via neural networks taking the parents' values as inputs (note that BNs are not restricted to discrete variables, so this perfectly defines a BN). It is not clear to me how to convert this BN into an SPN. Moreover, how would the authors convert a restricted Boltzmann machine (which is a MN) into an SPN? This will be hard, given the intractability of the RBM's partition function. Claiming equivalence between SPNs, BNs and MNs in the very introduction of the paper is a deepity, at best. The authors kindly invited me to re-read their paper, and re-iterated their contributions. However, the "what" was not the problem for me in the first place, but rather the "why". As said in my review, the use of hard EM for online structure learning is very promising and already challenging on its own. As illustrated above, there is also a lot of theory on this topic which should be addressed and build on. To additionally apply the framework to dynamic SPNs makes the paper rather convoluted and unfocused. Perhaps this is just personal opinion, but the authors did not explain their motivation in a satisfying manner. Due to the the shortcomings discussed above, I stick with my initial score. [1] Zhao et al., On the relationship between SPNs and BNs, 2015. [2] Poon and DOmingos, SPNs: a new deep architeccture, 2011. [3] Peharz et al., On the latent variable interpretation in SPNs, 2017. [4] Vergari et al., Sum-Product autoencoding, 2018. [5] Koller and Friedman, Probabilistic Graphical Models, 2009.